# Phenotypic Variation and Peel Contribution to Fruit Antioxidant Contents in European and Japanese Plums

**DOI:** 10.3390/plants11101338

**Published:** 2022-05-18

**Authors:** Pavlina Drogoudi, Georgios Pantelidis

**Affiliations:** Hellenic Agricultural Organization—‘DIMITRA’, Department of Deciduous Fruit Trees, Institute of Plant Breeding and Genetic Resources, 59035 Naoussa, Greece; gpantelidis76@gmail.com

**Keywords:** antioxidants, fruit color, fruit fresh weight, fruit shape, leaf characters, *Prunus domestica*, *Prunus salicina*, total soluble content

## Abstract

Herein, we studied the variation in leaf and fruit morphological traits and antioxidant contents in 43 local and foreign cultivars (cvs) grown under the same experimental conditions in the widely cultivated plum species *Prunus domestica* and *Prunus salicina*. The peel contribution of fruit bioactive compounds in a serving portion, correlations among the examined parameters, and group patterns in each plum species were also studied. The species and cvs were sufficiently separated. Compared to Japanese cvs, European cvs had less elongated leaves and smaller and sweeter fruit with less total phenol and antioxidant capacities. The Japanese cvs ‘Red ace’ and the widely grown ‘Black Amber’, together with the European ‘Tuleu Dulce’, ‘BlueFre’, and the landrace ‘Asvestochoriou’ make up groups with rich dietary sources of phytochemicals. The peel tissue contained higher total phenols and antioxidant capacities compared to the flesh, while the peel/flesh ratios varied widely among the cvs (6.6-fold). The variation in the antioxidant contents was lower among the cvs calculated per serving portion (3.7-fold); yet the peel tissue contribution was equal to that of the flesh (48.6%), signifying its high nutritive value. We observed increased sweetness in the fruit in the later-harvested cultivars, while cvs with more blue- and red-colored peel generally contained higher antioxidant contents mainly in the European plums. Moreover, larger fruit sizes were positively correlated with larger and more elliptic leaf shapes. In conclusion, the significant role of the genotype and the peel tissue as a source of bioactive compounds in plums were outlined with prospects of utilization in future breeding programs.

## 1. Introduction

Plums are of the most appreciated stone fruits worldwide, following peaches and nectarines in production (12.2 Mt of plums, 24.6 Mt of peach and nectarine; FAOSTAT 2020). Τhe most important commercial plum cultivars (cvs) belong to the hexaploid European plum (*Prunus domestica* L.) and the diploid Japanese plum (*Prunus salicina* Lindl.). Fresh fruit of *P. salicina* is mainly available during the summer and autumn months, while *P. domestica* is mostly processed into dried fruit and consumed year-round. Plum products such as jams and juice are produced from fruit that does not meet the fresh fruit marketable standards and is cherished year-round. Plums have a significant impact on human nutrition due to their health-promoting properties. They are also potentially valuable for the development and production of dietary supplements and functional food enriched with biologically active compounds [1]. The consumption of plum is related to osteoporosis prevention, improving memory, and has anti-inflammatory, antioxidant, and laxative effects [2].

The increased interest in research on plums is attributed to high levels of polyphenols that are known to be natural antioxidants; it is of interest to know the quantitative phenolics contained. More than 40 individual phenolics, mainly including chlorogenic acids (19), flavonols (9), flavan-3-ols (10), and anthocyanins (5), are accumulated in plum peel and flesh, with a remarkable diversity in their profiles depending on the variety and environmental conditions [1,3,4,5,6,7]. Moreover, the evaluation of the total antioxidant capacity, which takes into account the antioxidant activity of single compounds present in food as well as their potential synergistic interactions, is also of particular importance and has received much attention in recent years. In plums, the contribution of phenolic compounds to antioxidant activity was found to be more important than that of vitamin C and carotenoids [8].

Considering the significant impact of plum on human nutrition, it is of interest to consumers to define cvs with high polyphenolic content. There is high variability reported in the phytochemical and antioxidative compositions among different European [3,9,10,11,12,13,14,15] and Japanese plum cvs [16,17,18]. However, the above studies were conducted separately and little is known about the differences between those two widely cultivated species. The cultivation, climatic differences, and the ripening stage during harvest and postharvest treatments were also shown to affect the fruit’s phenolic and antioxidant contents [19,20].

Moreover, there is high inter-varietal phenotypic diversity in plum pomological traits, such as the ripening time, fruit shape and size, and peel and flesh color parameters [21]. Considering the above diversity, the European plum genotypes are classified into different pomological groups, including prunes, egg plums, greengages, damsons and bullaces, and mirabelle plums [21]. Peel color may be black, purple, red, green, or yellow, while flesh color may be yellow or red, with many shades of both colors; some cvs have a combination of both yellow and red flesh. The fruit color attributes are related to the anthocyanin contents [22,23]. Fruit color is also generally related to antioxidant content, but the use of peel color in predicting the fruit antioxidant content was reported to have limited application in fruit and vegetables with different colors in their inner and outer parts, such as in plums [23]. Relatively little is known about the potential contribution of the peel and flesh parts to the total antioxidant capacity in different plum cvs [8,11,18] or the relations between the fruit’s morphological traits and nutritional properties in plum [22,23,24,25,26].

European plum is native to Greece and local cultivars/accessions can be found abundantly. The local cvs with lower to higher commercial importance are: (a) ‘Skopelou’ (‘Agen Skopelou’ or ‘Glyka Skopelou’) on the island of Skopelos, Sporades regional unit, with significant production meant for drying, (b) ‘Mpardaki’ (in circular or elliptic shape) in the Fthiotida regional unit, and (c) ‘Praousti’ on the island of Samothrace, Evros regional unit. In the past, the local cv ‘Asvestochoriou’ also used to have a significant marketable share in central Macedonia. The pomological assignment and genetic relatedness of the local Greek cvs were previously characterized [21,27,28]. Due to the extensive use of foreign cvs with worldwide use, local germplasm is endangered. Therefore, the evaluation and exploitation of fruit quality and antioxidant attributes of Greek plum genetic material is of great importance and will assist in the promotion of interesting genotypes for breeding purposes to select cultivars with higher antioxidant content.

In the present study, the natural phenotypic variation in old and more recent commercial cultivars, as well as local cvs for the enhancement of plum antioxidant contents, were studied. Particularly, the objectives were to (a) determine the variations in the physical and chemical traits in fruit from 43 European and Japanese plum cvs grown under the same environmental and horticultural conditions, (b) examine the contribution to the fruit bioactive compounds in the peel and flesh tissue, and (c) dissect potential correlations among the examined physical and chemical traits.

## 2. Materials and Methods

### 2.1. Plant Material and Location

Fruits from 43 plum cvs; 19 European (*Prunus domestica*), 23 Japanese (*Prunus salicina* Lindl.), and 1 interspecific hybrid between plum and apricot (‘Pluot’) were analyzed (Table 1). From the European cvs, 12 were of international origin (1 from Bulgaria, 3 from France, 2 from Hungary, 2 from Romania, 1 from Russia, 1 from the United Kingdom, and 2 from the United States) and 7 were of local origin. The local cvs included ‘Skopelou’, ‘Asvestochoriou’, ‘Ksina Skopelou’ (also known as ‘Agiorgitiko’), ‘Avgato Skopelou’, ‘Mpardaki elliptic’, ‘Mpardaki circular’, and ‘Praousti’. Most of the studied Japanese plum cvs were of US origin (22), one was from Italy, and one from South Africa.

Fruits from the studied cvs were harvested from an evaluation orchard maintained at the Department of Deciduous Fruit Trees in Naoussa (40°37′ 13.40″ N; 22°06′ 59.80″ E, at 119 masl). The annual mean temperature was 15.7 °C. Precipitation was 717 mm, with lower values in the summer and higher values in the winter. Cultivars were grafted on GF677 rootstock and planted with a spacing of 5 × 4 m in soil with a medium-heavy mechanical composition and a neutral pH (pH 7.4). The trees were planted in a randomized block design of four trees per cultivar, with two replicate trees per block.

The fruit characteristics were measured on a 30-fruit sample per cultivar harvested at the commercial stage based on size, color, and firmness.

### 2.2. Fruit Morphometric and Colorimetric Traits

Upon harvesting, the fruit fresh weight (FW) was measured and the fruit shape was evaluated according to the UPOV characterization. Differentiation among the European plum cultivars was assigned based on the descriptions presented in the study by Gaši et al. (2020) to one of six pomological groups; (1) egg plums *sensu lato* (E), (2) prunes of the French d’Agen type (P/A), (3) prunes of the Central-Southeast European Zwetschen type (P/Z), (4) greengages (G), (5) mirabelles (M), and (6) bullaces, damsons and var. pomariorum (D/B) [21]. Differentiation between the two prune types is not clear-cut, but the P/A were described as having more pointed ends on both the fruits and stones.

The CIE color parameters *L** (brightness or lightness; 0 = black, 100 = white), *a** (−*a** = greenness, +*a** = redness) and *b** (−*b** = blueness, +*b** = yellowness), hue (h*) (calculated as tan^−1^*b**/*a**; 0° = red-purple, 90° = yellow, 180° = bluish-green, 270° = blue), and Chroma (C*) (calculated as (*a**^2^ + *b**^2^)^½^; degree of departure from grey to pure chromatic color) were measured in 12 fruit replicates, using a Minolta chromatometer (Minolta CR-400, Ramsey, NJ, USA). Readings were taken in the exocarp at both sides of each fruit and the mesocap was taken after cutting the fruit into slices.

The soluble solid content (SSC) and total acidity (TA) were determined from juice extracted using a food processor in three replicates of four plums. The SSC was determined using a digital refractometer (model PR-1, Atago, Japan) and expressed as °Brix, and the TA was analyzed in the juices by titration with 0.1 N NaOH to a pH endpoint of 8.2 and expressed as the citric acid equivalent (g/100 mL). The maturity index (MI) was calculated as the SSC/TA ratio.

### 2.3. Polyphenol Determinations and Antioxidant Capacities

Two wedge-shaped slices from the intact fruit were dissected, and the exocarp was separated and immediately frozen into liquid nitrogen and stored at −20 °C until needed. Frozen samples (about 0.5 g peel and 1.0 g flesh) were homogenized in 8 mL of 80% MeOH/H_2_O (*v*/*v*) using a mortar and pestle. The extract was centrifuged at 10,000× *g* for 10 min, and the supernatant was recovered.

#### 2.3.1. Total Phenols (TPs)

The TP content was measured using the Folin–Ciocalteu colorimetric method [29]. The reaction mixture consisted of 0.3 mL of diluted extract, 0.2 mL of distilled water, and 2.5 mL of 10% Folin–Ciocalteu reagent. The tube was vortexed and then allowed to stand at room temperature for 3 min while 2 mL of saturated sodium carbonate solution was added. The solution was incubated for 5 min at 50 °C, and the absorbance was measured at 760 nm against a blank solution. Each measurement was repeated in duplicate. The total phenolic content was expressed as mg gallic acid equivalents (GAE)/100 g FW.

#### 2.3.2. Total Antioxidant Capacity (TAC)

TAC was evaluated using 1,1- diphenyl-2-picrylhydrazyl (DPPH) (TAC_DPPH_) and ferric reducing antioxidant power (FRAP) (TAC_FRAP_) assays, and they were performed as described by Drogoudi et al. [30].

For the DPPH assay, the reaction mixtures containing 0 or 20 μL of diluted MeOH extract, 2.3 mL of 106.5 μM DPPH in MeOH, and 680 μL of H_2_O were vortexed and then kept at room temperature in the darkness for 4 h [31]. The absorbance of each reaction mixture was measured at 517 nm.

For the FRAP assay, a sample containing 3 mL of freshly prepared FRAP solution (0.3 Μ acetate buffer (pH 3.6) containing 10 mΜ 2,4,6-tripyridyl-s-triazine and 40 mΜ FeCl_3_ 10H_2_O) and 20 μL of peel or 50 μL of flesh extract was incubated at 37 °C for 4 min, and the absorbance was measured at 593 nm.

A standard curve was obtained on each measurement day, using ascorbic acid standard solution, and accordingly, the results are expressed as milligram ascorbic acid equivalents (AAE)/100g FW in peel (TAC_DPPH_–peel and TAC_FRAP_–peel) and flesh (TAC_DPPH_–flesh and TAC_FRAP_–flesh) tissue.

The TPs and antioxidant contents were also calculated per serving portion, with 100 g FW consisting of 5.8 g peel and 89.5 g flesh. This was based on a significant positive correlation (*r* = 0.840) found between the TPs measured in the total fruit vs. the TPs estimated from peel and flesh measurements in 12 European plum cvs [11]. A similar peel and flesh fresh weight contribution was found in the Japanese plum cv ‘Angeleno’ in the present study (data not shown). However, a higher peel contribution was used in the study of Gil et al. (15 g peel + 80 g flesh) [8].

### 2.4. Leaf Characteristics

The leaf characteristics, such as leaf shape, leaf tip shape, and the shape of the base, were evaluated according to UPOV. The leaf blade length (LBL), leaf blade width (LBW), and stalk length (SL) were measured. The ratios LBL/LBW and LBL/SL were calculated. The measurements were made in 12 fully developed leaves collected from the middle part of the previous year’s grown shoots.

### 2.5. Statistical Analyses

The data were subject to one-way analysis of variance (ANOVA), with cultivars and species as the treatments. The least significant difference (LSD) values were calculated in cases where significance at *p* < 0.05 variance was found. A correlation analysis was performed. Statistical analyses were performed using SPSS 13.0 (SPSS Inc., Chicago, IL, USA).

Principal component analysis (PCA) was applied to the mean values of the measured traits and a heatmap was created using the ClustVis online software [32]. Both the rows and columns were clustered using Euclidean distance and the Ward method.

## 3. Results and Discussion

### 3.1. Fruit Morphological Traits

Fruit from the European cvs was considerably diversified in size, shape, and color, and the cvs belonged to different pomological groups (Table 1 and Table 2; Appendix A). The landraces ‘Asvestohoriou’ and ‘Avgato Skopelou’ were classified as egg plums *sensu lato*, with a medium to large size and ovate or elliptic shape with rounded-end fruits and tender sweet flesh that often clings to the stone [21]. Anna Spath Pitesti’, ‘Mpardaki Circular’, ‘Prune d’ente 633’, and ‘Reine-Claude di Violette’ were classified as greengages, with a medium size, rounded and usually greenish fruits with tender and very sweet cling-stone flesh. ‘Praousti’ and ‘Russian’ were classified as mirabelles, with small, rounded fruits with yellow to orange peels and very sweet, free-stone flesh. ‘Ksina Skopelou’ and ‘Mpardaki Elliptic’ were classified as damsons, bullaces, or var. pomariorum. Finally, the Greek ‘Skopelou’ together with the remaining eight studied cvs were classified as prunes of the French d’Agen type or prunes of the Central-Southeast European Zwetschen type. ‘Skopelou’ is reported to be a sort of ‘d’Ente’; however, it does not have the characteristic pointed ends that refer to the French d’Agen type and is not considered to be self-fertile.

Among the Japanese plum cvs studied, ‘Pluot’, being an interspecific hybrid between Japanese plum and apricot, had the earliest ripening time (2 July), and after three days, ‘Beauty’, ‘Black Beauty’ and ‘Shiro’ ripened (Figure 1b). The ripening time of ‘Pluot’, in combination with its relatively high SSC and MI for the respective period (Figure 1a,b), provides an additional economic value for ‘Pluot’ cultivation. The earliest ripening European plum cvs were ‘Praousti’, ‘Asvestochoriou’, and ‘Russian’ with medium, large, and small fruit sizes, respectively, according to the classification described by Gaši et al. [21]. The superior features of ‘Asvestochoriou’ (large size and early ripening time) are the reasons it had a marketable value in Northern Greece in the past, but damages due to the spread of plum pox virus resulted in abandoning its cultivation.

There were no significant correlations between the ripening dates and the TPs or antioxidant capacities measured in the present study. Similar results were also found in the study by Arion et al. (2014), although they concluded that autumn compared to summer-harvested plum cvs contained higher phenolic content [33]. Nevertheless, in other fruit species, higher antioxidant contents were found in mid- to late-ripening cvs compared to earlier ripening cvs [30,34,35,36], which was related to higher temperatures and light intensity [37]. Moreover, precipitation during July–September was highly positively correlated with TPs in the fruit from Italian autochthonous plum cvs [15].

A large fruit size is preferred for both direct consumption and drying. Among the European plum cvs, extremely high fruit weights were found in ‘Prune d’ente 633’ and ‘Anna Spath Pitesti (70.5–101.7 g), and among the Japanese cvs, large or extremely large fruit sizes were found in ‘Autumn Giant’, ‘John W’, and ‘Angelino’ (120.9–151.0 g) (Table 1 and Table 2). The smallest FW was found in ‘Mpardaki elliptic’, ‘Russian’ (European cvs), ‘Shiro’, ’Beauty’, and ‘Laroda’ (Japanese cvs), being of a less favorable agronomic trait.

The greatest variations among all the studied morphological and chemical parameters were found in the peel blueness to yellowness (*b**–peel) (CV% = 204–262) and the flesh greenness to redness attributes (*a**–flesh) (CV% = 132–175) (Table 2; Appendix A). Among the European cvs, the most blue-colored peels (low *b** values) were found in ‘Stanley’, ‘Anna Spath Pitesti’, and ‘Scoldus SS’, while the most yellow (high *b** values) were in ‘Mpardaki elliptic’, ‘Praousti’, and ‘Mpardaki circular’ (Appendix A). The flesh color varied from being the reddest in cv ‘Russian’, followed by ‘D’ente 633’, to the most green in ‘Skopelou’ and ‘Anna Spath Oradea’, and yellow-colored in ‘Ksina Skopelou’. The plum ‘Russian’ was the only one among the European ones with a red-colored peel and flesh, suggesting it contains high anthocyanin contents. Concerning the Japanese plum cvs, the peel colors varied from the most yellow (‘Shiro’, ‘Sun Gold’, and ‘T.C. Sun’, high *b**–peel) to red (‘Fortune’, ‘Autumn Giant’, ‘Beauty’, ‘Ozark Premier’, and ‘Casselman’, highest *a**–peel). The collection contained only one genotype with red flesh, cv ‘Frontier’ (highest *a**–flesh = 23.8), while ‘Red ace’ and ‘Black Beauty’ also had reddish-colored flesh (Appendix A).

Cvs with more blue- and red-colored peel generally contained higher antioxidant contents mainly in European plum cvs (peel–b* vs. TPs–peel and TPs–serving, *r* = −0.546–(−0.592), peel–a* vs. TPs–peel, *r* = 0.577; Chroma–peel vs. TPs–peel, *r* = −0.568) although the correlations were not strong. Less significant correlations between color parameters and fruit antioxidants were found in Japanese plums (L–peel vs. TPs–peel or TAC_FRAP_–peel, *r* = −0.385–(−0.485)). Johnson et al. [26] and Yu et al. [25] observed the highest levels of phenolic compounds and antioxidant capacities in the black- or purple-flesh genotypes, although there were no significant correlations. In the study by Goldner et al. [24], conducted with the juice from 43 European plum varieties with yellow, blue, and dark blue fruit peels, weak red pigmentation co-occurred with low total phenol levels; however, there were exceptions, suggesting that the breeder can combine yellow fruit skin with a high level of health beneficial phenolic compounds by using the appropriate donor genotypes.

The b* color parameter (suggesting blue to yellow coloration) was positively correlated with its flesh counterpart (*r* = 0.678–0.736 in European and Japanese cvs) (Table 3 and Table 4). Only in European plum cvs, were the color parameters a*, hue, and Chroma also positively correlated in the peel and flesh (*r*= 0.464, −0.490, and 0.475, respectively). Although the color parameters appeared to correlate between the peel and flesh, there were no similar correlations in the TPs and antioxidant contents found in the peel and flesh (Table 3 and Table 4). Furthermore, the coloration and antioxidant contents were not previously correlated with their flesh counterparts in previous studies for apple [38] and pomegranate [39].

### 3.2. SSC and TA

European plums with SSC of more than 16 °Brix are considered sweet and suitable for fresh consumption [40]. In the present study, the SSC ranged from 11.5 to 19.6 °Brix in European plum cvs; only seven cvs had SSC higher than 16 °Brix, and ‘Skopelou’, ‘Tuley Dulce’, and ‘Stanley’ contained the highest values (mean of 19.1 °Brix) (Figure 1). Sahamishirazi et al. [13] found a wider range of SSC in a relatively high number of European plum cvs grown in Germany (178 cvs) (9.6–29.5%). In the present study, the range of SSC found in the Japanese was higher compared to the European plum cvs studied and ranged from 8.5 to 20.5 °Brix, with only four cvs having lower than 12 °Brix (‘Shiro’, ‘Black Beauty’, ‘Beauty’, and ‘Friar’).

The maturity index (MI) is the ratio between the SSC and TA (SSC/TA) contents that represents a reliable indicator of a cultivar’s suitability for acceptance by consumers [41]. The variation in the MI was higher than in the SSC (CV% = 50 and 17, respectively). The MI varied between 6.1 and 35.1, with the highest values in European plums ‘Stanley’, Tuley Dulce’, ‘Scoldus SS’, and ‘Skopelou’. Among the Japanese plum cvs ‘October Sun’, ‘Sun gold’, and ‘Pluot’ had the highest SSC and MI (means of 18.6 °Brix and 1.7, respectively). The earliest ripening cvs, ‘Shiro’, ‘Black Beauty’, and ‘Beauty’, had the lowest SSC and MI (means of 9.1 °Brix and 0.7, respectively) (Figure 1). Fruit from the cv ‘Shiro’ was also shown to have a low acceptance level in the study by Myracle et al. [42].

In summary, the European plums were smaller in size and had higher SSCs and MIs compared to the Japanese cultivars, whereas there was no difference in the TA. Higher SSCs in European plums were also found in the studies by Wolf et al. [18] and Liverani et al. [43], which further signifies the results obtained in the present study and suggests that European plums are more tasteful.

Fruit from the later-harvested plum cvs tended to be sweeter; the ripening date was positively correlated with the MI (*r* = 0.643) in the Japanese cvs and negatively correlated with the TA (*r*= −0.518) in the European plums (Figure 1), probably resulting from a greater fruit developmental time and enabling the fruit to become sweeter over a longer period. Previous studies have also reported that early-season cultivars had lower SSCs than late-season cultivars, while the TA was not related to the time of the season in plums [41] and other stone fruits [34,44,45,46,47].

### 3.3. TP and Radical Scavenging Activities

The TPs–peel content was lower in the European plum cvs (ranging from 152.1 to 984.6 mg/100 g FW) compared to the Japanese plums (202.7 to 1797.3 mg/100 g FW) (Table 2; Figure 2). The highest TPs–peel values were found in the European ‘Asvestochoriou’ and Skopelou’ and in the Japanese plums ‘Black Amber’, ‘Red Ace’, and ‘Florentia’. Relatively few studies have conducted separate measurements of TPs in the peel and flesh tissue, and similar or lower values have been reported for European (250–773 mg GAE/100 g FW in Cosmulescu et al. [11]) and Japanese plum cvs (72–656 in Wolf et al. [18]; 163.3–332.3 in Gil et al. [8]).

The TPs–flesh content was much lower compared to that in the peels of the European (55.9–220.9 mg GAE/100 g FW) and Japanese plum cvs (63.3–352.3 mg GAE/100 g FW). The highest values of TPs–flesh content were measured in the European ‘Tuley Dulce’, ‘Bluefre’, and ‘Asvestochoriou’ and the Japanese cvs ‘Red Ace’ and ‘Black Amber’. Similar values of TPs–flesh content have been found in other European (61–181 in Cosmulescu et al. [11]) and Japanese plum cvs (38–313 in Wolf et al. [18]; 22.0–76.9 in Gil et al. [8]).

To evaluate the impact of plum fruit consumption on the dietary intake of TPs and the ingested antioxidant equivalents, the antioxidants supplied by a plum fruit serving were determined. Among the 19 European plum cvs studied, the TPs–serving varied 3.7-fold (64.5–238.8 mg/100 g FW), and higher values were found in ‘Tuleu Dulce’, ‘BlueFre’, and ‘Asvestochoriou’, with the lowest in ‘Mpardaki Elliptic’, ‘Mpardaki circular’, and ‘Avgato’ Skopelou’ (Table 2; Figure 3). The local cvs ‘Skopelou’ and ‘Ksina Skopelou’ were also in the highest range of TPs–serving, and similar mean values were previously recorded [12].

Fruits from the Japanese cvs contained higher TPs–serving compared to European plums (an increase by 1.3-fold) and varied among the studied cvs by 4.6-fold (88.1–409.2 mg/100 g FW) (Table 2, Figure 3). Higher values of TPs–serving were found in ‘Red ace’, ‘Black Amber’, and ‘Angeleno’. Similarly, fruit from the cv ‘Black Amber’ was previously separated for having a high TP and antioxidant capacity when compared to other Japanese plum cvs (5 cvs, Lozano et al. [48]; 17 cvs, Wolf et al. [18]). Moreover, Gil et al. [8] similarly found high TPs and antioxidant content in ‘Angeleno’ fruit compared to four other cvs. In the present study, the lowest values of TPs–serving were found in ‘Laroda’, ‘T.C. Sun’, and ‘Santa Rosa’. The range of TPs found was similar to those reported for other Japanese plum cvs (227–383 mg GAE/100 g FW in 10 cvs, Venter et al. [16]; 317–761 in 5 yellow flesh genotypes, DiNardo et al. [17]; 51–430 in 15 cvs, Wolf et al. [18]).

The total antioxidant capacities were measured using the DPPH and FRAP methods, which determine the ability of antioxidants to reduce the DPPH or ferric complex (Fe^3+^) to the ferrous complex (Fe^2+^) via color and absorbance change, respectively. The TAC_DPPH_–peel and –flesh were similarly higher in the Japanese compared to the European plum cvs when measured in the peel, flesh, or per serving portion (Table 2). In the European plums, the mean values of the TAC_DPPH_–peel and –flesh were 594.1 and 121.9 mg AAE/100 g FW, respectively, while in the Japanese plums, they were 852.7 and 168.7 mg AAE/100 g FW, respectively (Figure 2). The above values are similar to those reported in the study by Gil et al. [8]. The fruit TAC_FRAP_–serving portion ranged from 56.0 to 154.6 mg AAE/100 g FW, which is similar to the ranges reported in the studies by Vasantha Rupasinghe et al. [10] and Dowling et al. [49] (105–424 and 121 to 229 mg, respectively), but significantly lower than the values reported by DiNardo et al. [17] (15,605 to 28,413 mg AAE/100 g FW).

The TP content was highly positively correlated with TAC_DPPH_ or TAC_FRAP_ in different plant tissues in the European and Japanese cvs (Table 3 and Table 4, respectively), with correlation *r* values usually being higher in the TAC_DPPH_ vs. TPs (ranging from 0.797 to 0.923) than in the TAC_FRAP_ vs. TPs (0.672–0.943, while it was non-significant in the peel tissue of the European cvs). Similarly, highly significant correlations have been found between the TPs and TAC_DPPH_ or TAC_FRAP_ in other studies [10,11,13,18]. The results from the present study suggest that measurements of TAC_DPPH_ could be better used than those of TAC_FRAP_ as a proxy for estimating the total phenolic content in plums, or vice versa, in screening programs.

### 3.4. Contribution of Peel to TPs and TAC in a Serving Portion

A large variation in the peel/flesh TP ratio in a cultivar (varied 6.6-fold, from 2.2 to 14.6 times) and peel/flesh TAC_DPPH_ and TAC_FRAP_ ratios (varied 9.6-fold, from 1.9 to 18.3 times) were found among the studied cvs. The peel tissue had a much higher content compared to its flesh counterpart in ‘Anna Spath Pitesti’ (European), ‘Florentia’, and ‘Black Star’ (Japanese cvs) (Figure 3). Nevertheless, lower ranges of peel/flesh TP ratios have been found in previous studies (4.5-fold in 12 European plums, Cosmulescu et al. [11]; 2.0-fold in Japanese plums, apart from ‘Shiro’, in which it was higher in the flesh, Wolf et al. [18]; 4.0–8.2-fold in 5 cvs, Gil et al. [8]), which may be related to the lower number of cvs studied.

Due to the low fruit weight contribution of the peel in a plum, the above large variation in the peel/flesh antioxidant contents found was diminished to a much lower variation when calculating the peel/flesh TPs and TAC_DPPH_ and TAC_FRAP_ ratios in a serving portion. This varied 3.7-fold from 12.3% to 48.6% (Figure 3). The percentage of the peel TP contribution was lower in the European (22.4%) compared to the Japanese (27.3%) plum cvs (Table 2) and coincided with having lower values of TP and antioxidant capacity per serving portion. The present study signifies that the peel tissue can have up to an equal contribution (48.6%) of the antioxidant contents contained in a serving portion as that of the flesh, and there is a relatively low variability among cvs.

### 3.5. Leaf Morphological Traits

In most of the studied cvs, the leaf shape was elliptic. The European and Japanese plum cvs differed in the leaf tip shape and shape of the leaf base, both being mainly acute in the Japanese cvs (Table 2; Appendix A). The eaves from the studied Japanese cvs were more elongated with shorter stalks (longer LBL, shorter LBW and SL, and there was a trend of having a higher LBL/LBW ratio, (*p* = 0.107)), compared to the European plum cvs. More elongated leaves (elliptic shape and higher LBL/LBW ratio) and higher LBL/SL ratios were found in ‘Asvestochoriou’ and ‘Anna Spath Pitesti’, whereas the smallest leaves and petioles were found in ‘Praousti’ (Appendix A). Significant positive correlations were found between the antioxidant contents in the fruit tissue and leaf shape (LBL/LBW) (*r* = 0.347–0.547). LBL and LBL/SL were positively correlated with FW (*r* = 0.376 and 0.550, respectively), suggesting that cvs with larger and more elliptic leaf shapes also produced larger fruit sizes. This was also reported in the study by Mirheidari et al. [50] that was conducted on autochthonous European plum accession in Iran; however, the *r* values were even lower (*r* < 0.300) compared to the present study.

### 3.6. Principal Component Analyses

In the present study, PCA was applied to a fruit physical and chemical dataset to determine the most important variables that explain the correlations between cultivars and to identify group patterns in each plum species separately. Two principal components explained a similar cumulative variation of 52.3–52.8% in both the European and Japanese plum cvs, presented as a scatter diagram in Figure 4b,d. In European plum cvs, the parameters with significant correlation to PC1 (values > 0.50) (34.7% of variance) were TPs and TAC (in the peel, flesh, and serving portion), SSC, and fruit physical traits, such as FW and color parameters (a*–peel, L–peel, and b*–peel); the local cvs ‘Avgato Skopelou’, ‘Mpardaki circular’, and ‘Mpardaki elliptic’ were separated from the rest of the studied cvs (Figure 4a).

In the Japanese plum cvs, the most important parameters incorporated into PC1 were the total phenol and antioxidant contents (in the peel, flesh, and serving portion) (36.1% of variance); ‘Black Amber’, ‘Red Ace’, and ‘Angeleno’ were separated with high values. PC2 (18.1% of variance) was positively correlated with the ripening date, L–peel, SSC, and MI and negatively correlated with the TA; the cvs ‘October Sun’, ‘Sun Gold’, ‘John W’ and T.C. Sun’ had high values (Figure 4c).

The collection of 19 European plum cultivars was classified into five clades (Figure 5). The first cluster included cvs Praousti’, ‘Avgato Skopelou’, ‘Mpardaki circular’, and ‘Mpardaki elliptic’, with more light and yellow-colored peel and flesh, high TA (high values in *b**–flesh, *L**–, *b**–peel; TA), and low antioxidant contents in the peel, flesh, and serving portion. Previously, results from a Bayesian structure analysis conducted using SSR–HRM markers on the same cvs showed that ‘Avgato Skopelou’ differed from the rest of the Greek plum cultivars since it was not grouped into the same cluster (Merkouropoulos et al., 2016) [27]. However, in the present study, ‘Avgato Skopelou’ was closely grouped with the other Greek cvs. The second cluster included the cvs ‘Skopelou’, ‘Tuley dulce’, ‘Asvestochoriou’, and ‘Bluefre’, with high fruit antioxidant contents. The third cluster included only the cv ‘Russian’, separated for red coloration in the peel and flesh, high TA, and low SSC and FW. The fourth cluster included the cvs ‘Giley’, ‘President’, ‘Ksina Skopelou’, ‘Anna Spath Oradea’, and ‘Reine-Claude di Violette’. Finally, the fifth cluster included the cvs ‘Anna Spath Pitesti’ (low L*–flesh), ‘Scoldus SS’, ‘Stanley’, ‘D’ente 632’, and ‘D’ente 633’, having low TA and darker-colored peel.

The collection of 24 Japanese plum cultivars was classified into seven clades (Figure 6). The first clade included ‘Angelino’, ‘Black amber’, and ‘Red Ace’, with high antioxidant contents. The second clade included the cvs ‘Shiro’, ‘Sun Gold’, ‘October Sun’, and ‘T.C. Sun’, with low antioxidant contents and high L– and b–peel. The third cluster included the cvs ‘Black Beauty’ and ‘Florentia’, with low L– peel and L–flesh. The fourth cluster included the cvs ‘Autumn Giant’, ‘John W’, ‘Frontier’, ‘Black Gold’, and ‘Pluot’. The fifth clade included the cvs ‘Laroda’, ‘Casselman’, and ‘Santa ‘Rosa’ (low antioxidant contents). The sixth cluster included the cvs ‘Calita’, ‘Simka’, ‘Black Star’, and ‘Friar’. Finally, the last cluster included the cvs ‘Beauty’, ‘Fortune’, and ‘Ozark Premier’.

## 4. Conclusions

The results from the present study give insight into the variability in leaf and fruit quality traits of 43 plum cvs from the most widely cultivated species *P. domestica* (European) and *P. salicina* (Japanese). The fruit was harvested under the same experimental conditions, and thus, the differences observed relate to genetic variability. To our knowledge, the above two species were not previously compared under the same experimental conditions. There were large genotype-dependent differences in the fruit qualitative traits and concentrations of the bioactive compounds studied; fruits from European plums were grouped as smaller and sweeter, with less phenol content and antioxidant capacity (1.3–1.6-fold decrease) compared to Japanese cvs.

The Japanese plums ‘Red ace’ and ‘Black Amber’ and the European plums ‘Tuleu Dulce’, ‘BlueFre’, and ‘Asvestochoriou’ were characterized by particularly rich dietary sources of phytochemicals. The highest TP content per serving portion was 409.2 mg in the cvs ‘Red ace’ and ‘Black Amber’, the latter being worldwide cultivated, and a value that is similar or higher to those reported for *Rubus idaeus, Ribes nigrum*, and *Sambucus nigra*, which are considered functional fruit [18]. This also contributes to the finding that plums are a rich source of phenolic compounds, being ranked superior to various other fruit, including those commonly consumed such as apple, banana, peach, pear, and watermelon [23,51]. Therefore, it is concluded that plums should be considered as having a significant impact on human nutrition, especially since they are widely consumed.

There was a large variation in the peel/flesh antioxidant content ratios found among the studied cvs, which was diminished to a much lower variation when expressed per serving portion. However, the peel contribution to the antioxidant content in a serving portion was almost equal (48.6%) in a cultivar, signifying that the peel can have significance equal to that of flesh in providing nutritive value in a plum fruit.

## Figures and Tables

**Figure 1 plants-11-01338-f001:**
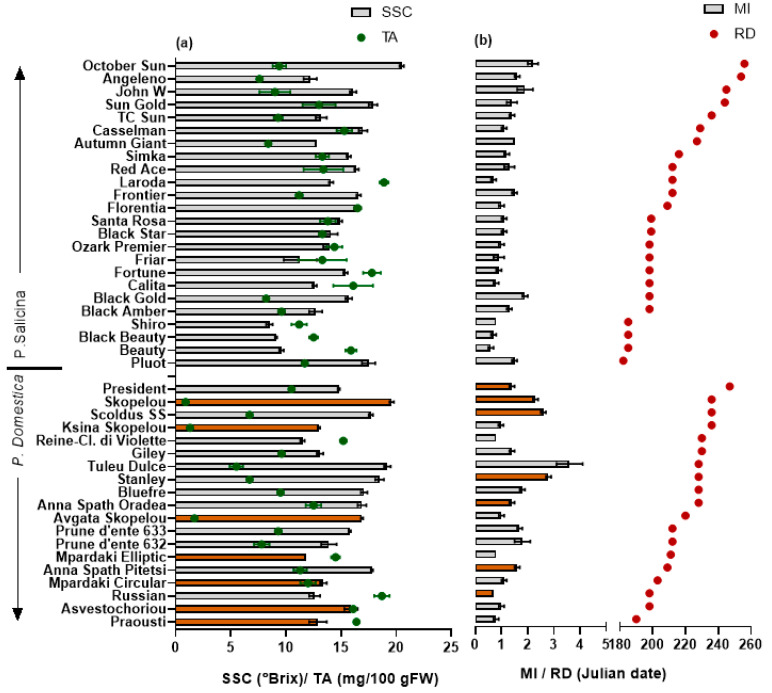
Mean (±SE) (**a**) soluble solid content (SSC, °Brix), titratable acidity (TA, mg citric acid/100 g FW), (**b**) maturity index (MI = SSC/TA), and ripening date (RD, Julian date) of 43 European and Japanese plums. Colored columns represent local cultivars. Least significant difference; SSC = 1.23, TA = 2.18, SSC/TA = 0.5.

**Figure 2 plants-11-01338-f002:**
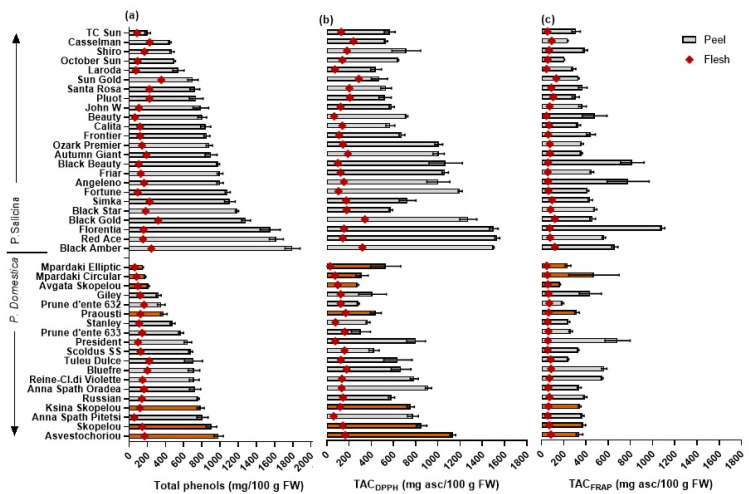
Mean (±SE) (**a**) total phenol content (mg gallic acid equivalent/100 g FW), (**b**) total antioxidant capacity using the DPPH (TAC_DPPH_), and (**c**) the FRAP radical (TAC_FRAP_) (mg ascorbic acid equivalent/100 g FW), in fruit peel (solid bars) and flesh (rhombus) tissue of 43 European and Japanese plum cvs. Colored columns represent local cultivars. LSD: TPs–peel, 143.5; TPs–flesh, 40.5; TAC_DPPH_–peel, 181.1; TAC_DPPH_–flesh, 38.3; TAC_FRAP_–peel, 172.4; TAC_FRAP_–flesh, 15.9.

**Figure 3 plants-11-01338-f003:**
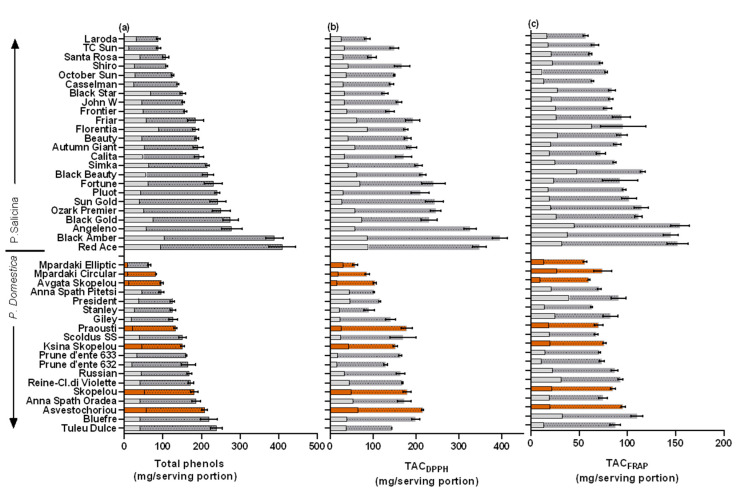
Mean (±SE) (**a**) total phenols (TPs, mg gallic acid equivalent) and total antioxidant capacity using the (**b**) DPPH (TAC_DPPH_) and (**c**) FRAP radicals (TAC_FRAP_) (mg ascorbic acid equivalent), expressed as per serving portion (100 g FW). The percentage (%) contributions of peel and flesh are shown as empty and hatched superimposed columns, respectively. Colored columns represent local cultivars. LSD; total phenols = 35.9, TAC_DPPH_ = 33.2, TAC_FRAP_ = 19.1.

**Figure 4 plants-11-01338-f004:**
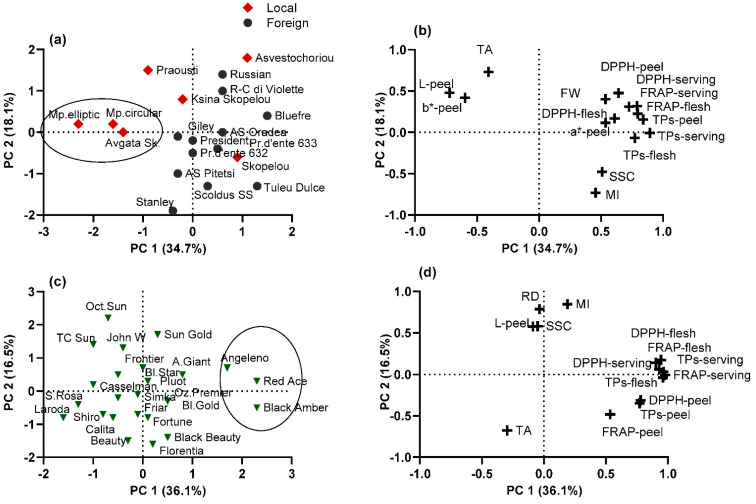
(**a**,**c**) Segregation and (**b**,**d**) factor loadings, of (**a**,**b**) 19 European and (**c**,**d**) 24 Japanese plum cultivars, on the basis of fruit physical and chemical characters, determined by principal component analysis. Variable annotations are presented in Table 3.

**Figure 5 plants-11-01338-f005:**
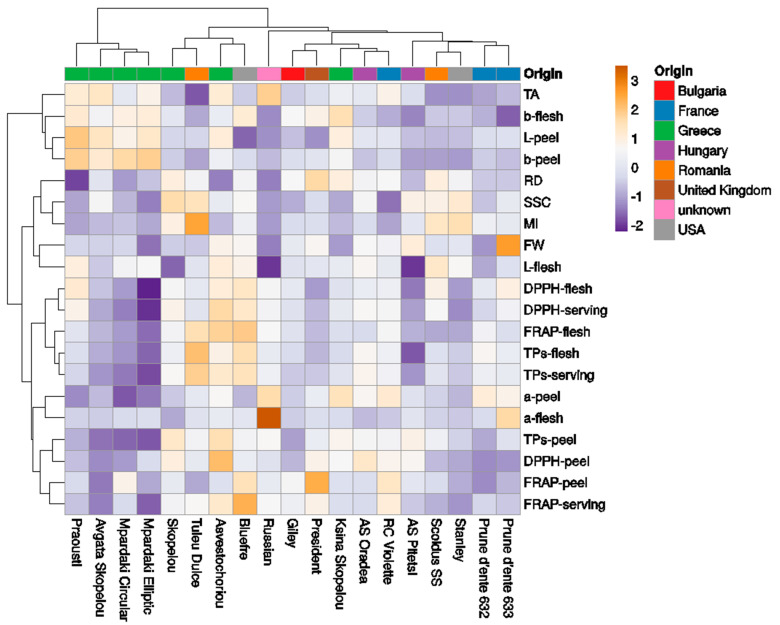
Heatmap showing the clustering of fruit phenotyping traits in 19 European plum cultivars using the ClustVis software. The columns correspond to the cultivars and the rows correspond to the fruit phenotypic traits studied. Both the rows and columns were clustered using Euclidean distance and the Ward method.

**Figure 6 plants-11-01338-f006:**
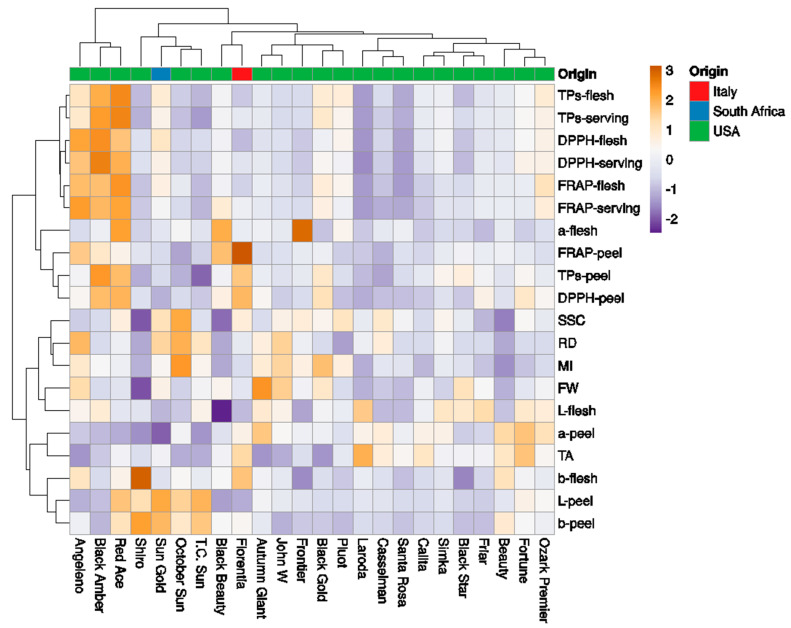
Heatmap showing the clustering of fruit phenotyping traits in 24 Japanese plum cultivars using the ClustVis software. The columns correspond to the cultivars and the rows correspond to the fruit phenotypic traits studied. Both rows and columns were clustered using Euclidean distance and the Ward method.

**Table 1 plants-11-01338-t001:** Cultivar names, origin, fruit fresh weight, and fruit shape in lateral view (UPOV characterization) of the European and Japanese plum cultivars studied.

	Origin	Fruit Weight (g)	Fruit Shape
European cvs			
Anna Spath Oradea	Hungary	61.1	Elliptic
Anna Spath Pitetsi	Hungary	70.5	Circular
Asvestochoriou	Greece	65.3	Circular
Avgata Skopelou	Greece	40.0	Drop shape
Bluefre	USA	62.9	Oblate
Giley	Bulgaria	51.5	Ovate
Ksina Skopelou ^1^	Greece	26.5	Obovate
Mpardaki Circular	Greece	41.3	Circular
Mpardaki Elliptic	Greece	18.4	Elliptic
Praousti	Greece	41.7	Ovate
President	UK	63.1	Oblong
Prune d’ente 632	France	25.6	Ovate
Prune d’ente 633	France	101.7	Circular
Reine-Cl.di Violette	France	57.5	Circular
Russian	Russia	20.7	Circular
Scoldus SS	Romania	45.5	Elliptic
Skopelou ^2^	Greece	38.5	Elliptic
Stanley	USA	49.1	Oblong
Tuleu Dulce	Romania	37.3	Ovate
Japanese cvs			
Angeleno	USA	120.9	Oblate
Autumn Giant	USA	151.0	Circular
Beauty	USA	53.8	Cordate
Black Amber	USA	68.2	Oblate
Black Beauty	USA	97.8	Oblate
Black Gold	USA	112.3	Oblate
Black Star	USA	116.2	Circular
Calita	USA	84.6	Circular
Casselman	USA	62.1	Obovate
Florentia	Italy	77.0	Cordate
Fortune	USA	78.1	Obovate
Friar	USA	92.4	Circular
Frontier	USA	87.7	Oblate
John W	USA	127.8	Circular
Laroda	USA	53.8	Circular
October Sun	USA	65.8	Oblong
Ozark Premier	USA	90.2	Oblate
Pluot	USA	69.8	Cordate
Red Ace	USA	75.5	Oblate
Santa Rosa	USA	57.5	Oblong
Shiro	USA	28.0	Circular
Simka	USA	64.7	Cordate
Sun Gold	South Africa	100.2	Obovate
T.C. Sun	USA	90.7	Circular

^1^ Also named ‘Agiorgitiko’; ^2^ also named ‘Agen Skopelou’ and Glyka Skopelou’.

**Table 2 plants-11-01338-t002:** Mean (minimum–maximum), percentage coefficient variation (CV%), and *p* values of fruit quality and leaf traits when compared among 19 European and 24 Japanese plum cultivars grown in a cultivar evaluation orchard in Naoussa, Greece. Parameters measured were fruit fresh weight (g), peel and flesh color CIELAB parameters, soluble solid content (SSC, °Brix), titratable acidity (TA, grams citric acid equivalent/100 mL), maturity index (SSC/TA), total phenols (TPs) (mg GAE/100 g FW), total antioxidant capacity using the DPPH and FRAP methods (TAC_DPPH_ and TAC_FRAP_, mg AAE/100 g FW) in peel and flesh tissue and serving portion (100 g FW), percentage TPs–peel/TPs–serving (% TPs–p/s), leaf shape (1, ovate; 2, elliptic; 3, obovate), leaf tip shape (1, acute; 2, right-angled; 3, obtuse), shape of base (1, acute; 2, obtuse; 3, truncate)**,** leaf blade length (LBL, cm), leaf blade width (LBW, cm), stalk length (SL, cm) and ratios of LBL/LBW and LBL/SL. Different letters in the line indicate significant differences.

	European			Japanese			
	Mean	Min–Max	CV%	Mean	Min–Max	CV%	*p*
Fruit weight	48.3 b	18.4–101.7	42	84.4 a	28.0–151.0	33	<0.001
*L*–peel	38.2	22.5–57.4	25	36.1	21.3–59.7	32	0.686
*a**–peel	9.8 b	−23.7	73	13.5 a	−31.2	63	0.009
*b**–peel	9.3	−56.6	204	5.3	−46.2	262	0.840
*h**–peel	162.0	38.8–319	162	186.6	20.3–337.0	67	0.324
Chroma–peel	21.3	6.7–45.0	57	21.0	9.4–36.2	262	0.221
*L*–flesh	50.5 b	37.4–59.0	13	55.3 a	38.6–66.1	12	<0.001
*a**–flesh	4.8	−37.3	175	5.0	−25.2	132	0.188
*b**–flesh	25.1 a	7.7–41.2	41	20.8 b	8.3–44.3	38	0.012
h*–flesh	77.9	19.2–97.5	27	77.1	20.5–95.3	22	0.197
Chroma–flesh	27.3 a	12.7–41.3	33	21.9 b	8.4–44.3	35	<0.001
SSC	15.4 a	11.5–19.6	17	14.4 b	8.5–20.5	20	0.046
TA	1.2	0.5–1.9	33	1.3	0.8–1.9	25	0.077
Maturity index	15.4 a	6.7–35.1	50	12.1 b	6.1–21.7	34	0.001
TPs–peel	582.5 b	152.1–984.6	43	916.7 a	202.7–1797.3	42	<0.001
TAC_DPPH_–peel	594.1 b	277.9–1135.7	42	852.7 a	443.9–1535.8	41	<0.001
TAC_FRAP_–peel	361.4 b	166.9–685.1	37	456.8 a	206.1–1090.8	44	0.008
TPs–flesh	130.1 b	55.9–220.9	33	164.5 a	63.3–352.3	45	0.006
TAC_DPPH_–flesh	121.9 b	30.8–178.5	34	168.7 a	66.9–343.9	42	<0.001
TAC_FRAP_–flesh	64.7 b	47.2–86.3	17	76.8 a	44.8–133.6	31	0.003
TPs–serving	150.2 b	64.5–238.8	31	200.4 a	88.1–409.2	41	<0.001
TAC_DPPH_–ser.	143.6 b	58.4–215.0	29	199.3 a	85.7–394.9	37	<0.001
TAC_FRAP_–ser.	78.4 b	56.0–109.9	17	94.3 a	56.9–154.6	29	0.001
% TPs–p/s	22.4	12.3–48.3		27.3	13.3–48.6		
Leaf shape	2.1	1–3	40	2.1	1–3	20	0.932
Leaf tip shape	1.9	1–3	43	1.3	1–3	48	0.005
Shape of base	1.6	1–3	37	1.0	1–3	20	<0.001
LBL	8.8 b	5.6–12.0	17	9.7 a	8.0–12.8	12	<0.001
LBW	5.3 a	3.5–6.9	18	4.2 b	3.0–5.6	17	<0.001
SL	1.7 a	1.1–2.6	23	1.4 b	1.0–2.2	23	<0.001
LBL/LBW	1.7	1.4–2.2	14	2.4	1.8–3.2	14	0.107
LBL/LPL	5.6 b	3.8–10.0	30	7.3 a	4.8–10.7	18	<0.001

**Table 3 plants-11-01338-t003:** Pearson correlation analyses between phenotypic and chemical traits in 19 European plum cvs. RD, ripening date; FW, fruit fresh weight; SSC, soluble solid content; TA, titratable acidity; MI, maturity index; TPs, total phenols; DPPH and FRAP, total antioxidant capacity using the DPPH and FRAP radicals, respectively. ns. non significant; Absolute linear correlations ≥|0.60| are marked in bold.

	1	2	3	4	5	6	7	8	9	10	11	12	13	14	15	16	17	18	19	20	21	22	23	24
1. RD	1																							
2. FW	ns	1																						
3. *L**−peel	−0.473	ns	1																					
4. *a**−peel	ns	ns	ns	1																				
5. *b**−peel	ns	ns	**0.793**	−0.492	1																			
6. *h**−peel	ns	ns	ns	ns	−**0.678**	1																		
7. Chroma−peel	−**0.607**	ns	0.844	ns	**0.888**	ns	1																	
8. *L**−flesh	ns	ns	ns	ns	ns	ns	ns	1																
9. *a**−flesh	ns	ns	ns	0.464	ns	ns	ns	ns	1															
10. *b**−flesh	ns	ns	ns	−0.471	**0.736**	−**0.845**	ns	ns	ns	1														
11. *h**−flesh	ns	ns	ns	−0.467	ns	−0.490	ns	ns	−**0.950**	**0.583**	1													
12. Chroma−flesh	ns	ns	ns	ns	**0.677**	−**0.675**	0.475	ns	ns	**0.806**	ns	1												
13. SSC	ns	ns	ns	ns	−0.546	ns	−0.571	ns	ns	ns	ns	−0.481	1											
14. TA	−0.518	ns	0.482	ns	0.575	ns	0.595	ns	ns	ns	ns	0.523	−0.598	1										
15. MI	ns	ns	ns	ns	−**0.625**	0.553	−0.578	ns	ns	ns	ns	−0.548	**0.784**	−**0.894**	1									
16. TPs−peel	ns	ns	ns	0.577	−0.592	ns	−0.568	ns	ns	ns	ns	ns	ns	ns	ns	1								
17. DPPH−peel	ns	ns	ns	ns	ns	ns	ns	ns	ns	ns	ns	ns	ns	ns	ns	**0.797**	1							
18. FRAP−peel	ns	ns	ns	ns	ns	ns	ns	ns	ns	ns	ns	ns	ns	ns	ns	ns	ns	1						
19. TPs−flesh	ns	ns	ns	ns	ns	ns	ns	ns	ns	ns	ns	ns	ns	ns	ns	ns	ns	ns	1					
20. DPPH−flesh	ns	ns	ns	ns	ns	ns	ns	ns	ns	ns	ns	ns	ns	ns	ns	0.469	ns	ns	**0.738**	1				
21. FRAP−flesh	ns	ns	ns	ns	ns	ns	ns	ns	ns	ns	ns	ns	ns	ns	ns	0.567	ns	ns	**0.872**	**0.702**	1			
22. TPs−serving	ns	ns	ns	0.490	−0.522	ns	−0.546	ns	ns	ns	ns	ns	ns	ns	ns	**0.684**	0.457	ns	**0.961**	**0.750**	**0.891**	1		
23. DPPH−serving	ns	ns	ns	ns	ns	ns	ns	ns	ns	ns	ns	ns	ns	ns	ns	**0.680**	0.477	ns	**0.719**	**0.942**	**0.754**	**0.801**	1	
24. FRAP−serving	ns	ns	−0.484	ns	ns	ns	−0.477	ns	ns	ns	ns	ns	ns	ns	ns	0.598	0.563	**0.684**	**0.637**	0.524	**0.818**	**0.708**	**0.650**	1

**Table 4 plants-11-01338-t004:** Pearson correlation analyses between phenotypic and chemical traits in 24 Japanese plum cvs. Abbreviations as in Table 3.

	1	2	3	4	5	6	7	8	9	10	11	12	13	14	15	16	17	18	19	20	21	22	23	24
1. RD	1																							
2. FW	ns	1																						
3. *L**−peel	ns	ns	1																					
4. *a**−peel	ns	ns	ns	1																				
5. *b**−peel	ns	ns	0.749	ns	1																			
6. *h**−peel	ns	ns	−0.414	ns	−**0.804**	1																		
7. Chroma−peel	ns	ns	**0.759**	ns	**0.773**	−**0.606**	1																	
8. *L**−flesh	ns	ns	ns	ns	ns	ns	ns	1																
9. *a**−flesh	ns	ns	ns	ns	ns	ns	ns	−**0.612**	1															
10. *b**−flesh	ns	ns	ns	ns	**0.678**	−**0.721**	0.552	ns	ns	1														
11. *h**−flesh	ns	ns	ns	ns	ns	ns	ns	0.575	−**0.963**	ns	1													
12. Chroma−flesh	ns	ns	ns	ns	0.583	−0.574	ns	ns	ns	**0.791**	−0.192	1												
13. SSC	0.488	ns	ns	ns	ns	ns	ns	ns	ns	ns	ns	ns	1											
14. TA	−0.421	−0.493	ns	ns	ns	ns	ns	ns	ns	ns	ns	ns	ns	1										
15. MI	**0.643**	0.438	ns	ns	ns	ns	ns	ns	ns	ns	ns	ns	**0.620**	−**0.774**	1									
16. TPs−peel	ns	ns	−0.385	ns	ns	ns	−0.409	ns	ns	ns	ns	ns	ns	ns	ns	1								
17. DPPH−peel	ns	ns	−0.181	ns	ns	ns	−0.132	ns	ns	ns	ns	ns	ns	ns	ns	**0.828**	1							
18. FRAP−peel	ns	ns	−0.495	ns	ns	ns	−0.334	ns	ns	ns	ns	0.411	ns	ns	ns	**0.672**	**0.683**	1						
19. TPs−flesh	ns	ns	ns	ns	ns	ns	ns	ns	ns	ns	ns	ns	ns	ns	ns	**0.629**	**0.609**	0.259	1					
20. DPPH−flesh	ns	ns	ns	ns	ns	ns	ns	ns	ns	ns	ns	ns	ns	ns	ns	0.511	0.511	0.291	**0.923**	1				
21. FRAP−flesh	ns	ns	ns	ns	ns	ns	ns	ns	ns	ns	ns	ns	ns	ns	ns	0.590	**0.623**	0.306	**0.943**	**0.702**	1			
22. TPs−serving	ns	ns	ns	ns	ns	ns	ns	ns	ns	ns	ns	ns	ns	ns	ns	**0.777**	**0.716**	0.39	**0.978**	**0.750**	**0.891**	1		
23. DPPH−serving	ns	ns	ns	ns	ns	ns	ns	ns	ns	ns	ns	ns	ns	ns	ns	**0.660**	**0.705**	0.411	**0.941**	**0.942**	**0.754**	**0.801**	1	
24. FRAP−serving	ns	ns	ns	ns	ns	ns	ns	ns	ns	ns	ns	ns	ns	ns	ns	**0.703**	**0.726**	0.565	**0.887**	0.524	**0.818**	**0.708**	**0.650**	1

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
