# Peer review of "Phenotypic Variation and Peel Contribution to Fruit Antioxidant Contents in European and Japanese Plums"

_plants, 2022, doi:10.3390/plants11101338_

Round 1

Reviewer 1 Report

Dear authors, please find my remarks and suggestions.

  1. Line 14: remove bold
  2. Change presentation of references in the text. According to the authors guidelines, references must be numbered in order of appearance in the text.
  3. Format line 117.
  4. Line 143. Please explain your choice to use SSC/TA ratio as maturity index or provide reference. This ratio is related to the taste and usually is used for consumer acceptance. Normally, maturity of plums is described by flesh firmness or colour development.
  5. Table 2. Add note that different letters in the line are indicating significant differences.
  6. Table 3. Title of the table is split by the table.
  7. Correct titles of tables 3 and 4. Phenol content, antioxidant activity, SSC or TA are not phenotypic traits.
  8. Lines 360-361. Not understandable.
  9. Leaf traits. I suggest to remove this chapter from the manuscript. It does not fit to other results, mostly presenting various fruit quality attributes. Also, it is rather strange to look and to find correlations between leaf shape and fruit phenol content.
  10. Diminish a number of references of your own papers. There should be minimum discussions with yourself. Do not use three references of your own publications for the same statement (line 241-242): “Significant but not strong correlations were found between harvest date and antioxidant contents in peach (Drogoudi et al., 2017; 2016; Drogoudi and Tsipouridis, 2007).” Check other references of your own papers.
  11. Figures 1 and 2 are complicated to analyze and to find significant differences between cultivars. Better to indicate differences by letters, not by providing LSD value. I suggest to present results of Fig.1 in a table. Do the same with Fig.2 or divide into two figures, presenting separately flesh and peel results. In this case, you can indicate significance by letters.

Author Response

We would like to thank you and the reviewers for your time, recommendations and positive remarks on the above submission for publication. In the revised manuscript the changes were made in a track change mode so that they can be easily recognized. Please find below the changes made and replies to the Reviewers, who made some very prompt remarks.

Reviewer #1

  • Line 14: remove bold DONE
  • Change presentation of references in the text. According to the author’s guidelines, references must be numbered in order of appearance in the text. DONE
  • Format line 117.
  • Line 143. Please explain your choice to use SSC/TA ratio as maturity index or provide reference. This ratio is related to the taste and usually is used for consumer acceptance. Normally, maturity of plums is described by flesh firmness or colour development.

In the section 3.2 SSC and TA and explanation and reference related to maturity index is written. Indeed, mostly changes in flesh firmness during ripening is the parameter related with ripening, yet in the present study we are interested in the parameters related with fruit quality as recognized by the consumers.

  • Table 2. Add note that different letters in the line are indicating significant differences. DONE
  • Table 3. Title of the table is split by the table. DONE
  • Correct titles of tables 3 and 4. Phenol content, antioxidant activity, SSC or TA are not phenotypic traits. DONE.
  • Lines 360-361. Not understandable. Line numbers did not show in the text we have, although it is indicated to be present. Nevertheless I counted all lines and I suppose it refers to the text ‘ There was no significant correlation between ripening date and TPs or antioxidant capacities measured in the present study, which may be related to the high variation in other morphometric characteristics among cvs’ The above sentence was changed to ‘There was no significant correlation between ripening date and TPs or antioxidant capacities measured in the present study’
  • Leaf traits. I suggest to remove this chapter from the manuscript. It does not fit to other results, mostly presenting various fruit quality attributes. Also, it is rather strange to look and to find correlations between leaf shape and fruit phenol content.

We were also puzzled whether or not to include the leaf morphological traits in the present study. Yet we decided to include them cause yet it is a new finding and similar results on correlations between leaf and fruit sizes was found in the study by Mirheidari et al. (2020)  

  • Diminish a number of references of your own papers. There should be minimum discussions with yourself. Do not use three references of your own publications for the same statement (line 241-242): “Significant but not strong correlations were found between harvest date and antioxidant contents in peach (Drogoudi et al., 2017; 2016; Drogoudi and Tsipouridis, 2007).” Check other references of your own papers. DONE
  • Figures 1 and 2 are complicated to analyze and to find significant differences between cultivars. Better to indicate differences by letters, not by providing LSD value. I suggest to present results of Fig.1 in a table. Do the same with Fig.2 or divide into two figures, presenting separately flesh and peel results. In this case, you can indicate significance by letters.

We also though of the suggested option of presenting data, yet we think that in the present form the reader can visualize the relative differences among species, cvs and fruit tissue in a relatively high number of cultivars, that would not be possible if presented in a table. Moreover, the high number of letter separating a large number of cvs would also be confusing to the reader.

Reviewer 2 Report

Dear authors,

this is an interesting article. however, it needs a bit of reformating as the paper design included the shape of fruits as one of the parameters of comparison between the European and Japanese species. Can you please clarify the scope of highlighting this parameter within this research paper? Also, I would like to suggest some linguistic notations e.g using the word "lower" would be better than writing "smaller " in line 81. Finally, wish you all the best in publishing this article. 

Author Response

Reviewer #2

this is an interesting article. however, it needs a bit of reformating as the paper design included the shape of fruits as one of the parameters of comparison between the European and Japanese species. Can you please clarify the scope of highlighting this parameter within this research paper?

Thank you for your suggestions. Fruit shape was presented as a mean to separate the cvs. The two species cannot be compared according to fruit shape cause the different shapes studied were not similar in the two species, as suggested by the UPOV characterization.

Also, I would like to suggest some linguistic notations e.g using the word "lower" would be better than writing "smaller " in line 81. Finally, wish you all the best in publishing this article. DONE

Round 2

Reviewer 1 Report

Thank you for your explanations.

it is not Table 2, but Fig.2.  Figure is not completely displayed. Part of this figure on the right side is hidden. 

Author Response

Dear Reviewer,

I shortened the horizontal dimension in Table 2 so that it fits better in the page, although I could not distinguish a problem. Perhaps the editorial office will spot any similar mistake and correct accordinetly if needed.

Thank you so much again for your time and useful comments.

Yours sincerely

Pavlina Drogoudi